# Buds and Bugs: A Fascinating Tale of Gut Microbiota and Cannabis in the Fight against Cancer

**DOI:** 10.3390/ijms25020872

**Published:** 2024-01-10

**Authors:** Ahmad K. Al-Khazaleh, Kayla Jaye, Dennis Chang, Gerald W. Münch, Deep Jyoti Bhuyan

**Affiliations:** 1NICM Health Research Institute, Western Sydney University, Penrith, NSW 2751, Australia; 19316068@student.westernsydney.edu.au (A.K.A.-K.); 19255718@student.westernsydney.edu.au (K.J.); d.chang@westernsydney.edu.au (D.C.); 2Pharmacology Unit, School of Medicine, Western Sydney University, Campbelltown, NSW 2560, Australia; 3School of Science, Western Sydney University, Penrith, NSW 2751, Australia

**Keywords:** gut microbiota, Cannabis, cancer, Cannabinoids, THC, CBD, gut microbiome, personalized medicine, microbiome-cannabis axis

## Abstract

Emerging research has revealed a complex bidirectional interaction between the gut microbiome and cannabis. Preclinical studies have demonstrated that the gut microbiota can significantly influence the pharmacological effects of cannabinoids. One notable finding is the ability of the gut microbiota to metabolise cannabinoids, including Δ^9^-tetrahydrocannabinol (THC). This metabolic transformation can alter the potency and duration of cannabinoid effects, potentially impacting their efficacy in cancer treatment. Additionally, the capacity of gut microbiota to activate cannabinoid receptors through the production of secondary bile acids underscores its role in directly influencing the pharmacological activity of cannabinoids. While the literature reveals promising avenues for leveraging the gut microbiome–cannabis axis in cancer therapy, several critical considerations must be accounted for. Firstly, the variability in gut microbiota composition among individuals presents a challenge in developing universal treatment strategies. The diversity in gut microbiota may lead to variations in cannabinoid metabolism and treatment responses, emphasising the need for personalised medicine approaches. The growing interest in understanding how the gut microbiome and cannabis may impact cancer has created a demand for up-to-date, comprehensive reviews to inform researchers and healthcare practitioners. This review provides a timely and invaluable resource by synthesizing the most recent research findings and spotlighting emerging trends. A thorough examination of the literature on the interplay between the gut microbiome and cannabis, specifically focusing on their potential implications for cancer, is presented in this review to devise innovative and effective therapeutic strategies for managing cancer.

## 1. Introduction

The gut microbiota is a dynamic and diverse community of microorganisms residing within the gastrointestinal tract, consisting of bacteria, archaea, viruses, and fungi. This complex ecosystem is fundamental in various physiological processes, including nutrient metabolism, immune regulation, and protection against pathogenic invaders [1,2]. Additionally, the gut microbiota is now recognised as a crucial modulator of drug metabolism and efficacy along with the liver, influencing the bioavailability and activity of numerous pharmaceutical agents [3,4,5,6,7,8].

*Cannabis sativa*, a plant from the Cannabis species, has been used for medicinal and recreational purposes for centuries. The therapeutic properties of cannabis are mainly attributed to its active compounds known as cannabinoids, with delta-9-tetrahydrocannabinol (THC) and cannabidiol (CBD) being the most extensively studied [9]. The endocannabinoid system (ECS) in the body, comprising cannabinoid receptors, endogenous ligands (endocannabinoids), and enzymes, plays a pivotal role in mediating the effects of cannabis on various physiological processes [10]. Recent research advancements have uncovered a bidirectional relationship between the gut microbiota and the ECS, underpinning the potential influence of gut microbiota on cannabinoid metabolism and vice versa [11,12]. Consequently, investigations into the interaction between gut microbiota and cannabis have gained significant attention, given its potential impact on human health and diseases.

This review aims to decipher the interaction between human gut microbiota and cannabis constituents, predominantly cannabinoids, and their potential implications in carcinogenesis, cancer progression, and therapeutic approaches. Firstly, it explores the existing literature highlighting the role of the gut microbiota in influencing the metabolism, absorption, and bioavailability of cannabinoids. Understanding these interactions offers insights into how specific gut microbial strains could modify the effects of cannabinoids, possibly impacting their anticancer properties or adverse effects. In addition, this study aims to uncover how cannabinoids, in return, can shape the composition and behaviour of the gut microbiota. Recent reviews have investigated the ability of the gut microbiota to mediate the effects of cannabinoids, including their metabolism and distribution throughout the body [13,14,15,16]. Furthermore, research has explored the bidirectional nature of this interplay, examining how cannabinoids might influence the makeup and functionality of the gut microbiota.

Through investigating whether distinct cannabinoids exhibit prebiotic or antimicrobial properties, studies have aimed to discern whether cannabis-derived compounds could foster advantageous gut microbial populations, consequently contributing to strategies for averting or treating cancer. While the precise mechanisms underpinning this interplay are still in the elucidation process, several investigations have commenced delineating the potential import of this interface, particularly in the context of cancer.

### 1.1. Cancer and Gut Microbiota

Cancer remains a leading cause of morbidity and mortality worldwide, necessitating continuous efforts to develop novel therapeutic strategies. Cancer also is a complex disease, and there is no single cause. However, some foods and dietary components may increase or decrease the risk of developing certain types of cancer [17,18,19]. Moreover, some of the carcinogens found in food include Acrylamide. This chemical is formed when starchy foods are cooked at high temperatures, such as frying, baking, or roasting. Acrylamide is found in French fries, potato chips, and coffee [17]. HCAs and PAHs are also formed when muscle meat, including beef, pork, fish, and poultry, is cooked using high-temperature methods such as grilling or broiling. Exposure to these chemicals has been linked to an increased risk of colorectal, pancreatic, and prostate cancers [20,21].

Furthermore, nitrites and nitrates chemicals are used as preservatives in processed meats such as bacon, ham, and hot dogs. When these meats are cooked at high temperatures, they can form nitrosamines, known as carcinogens [22,23]. It is important to note that while some foods and dietary components may increase the risk of cancer, there is no evidence that specific foods can cause or cure cancer [24,25]. A healthy diet that includes a variety of fruits, vegetables, whole grains, and lean proteins can help reduce the risk of cancer and other chronic diseases [24,25,26,27].

The interplay between gut microbiota and cancer has been an emerging area of investigation, unveiling the potential influence of microbial communities on the onset of cancer, progression, and therapeutic outcomes. Recent scientific endeavours have highlighted the consequences of perturbations in gut microbiota composition and diversity, commonly called dysbiosis. Such perturbations can contribute to chronic inflammation and undermine immune surveillance, ultimately promoting an environment conducive to the initiation and development of carcinogenesis [28,29]. Moreover, microbial metabolites arising from dietary substrates and gut fermentation have become critical contributors to cancer-related pathways. Short-chain fatty acids (SCFAs), including butyrate, acetate, and propionate, produced as a result of fibre fermentation in the gut, exert notable anti-inflammatory and antiproliferative effects, thus influencing the microenvironment of cancer cells and potentially curtailing cancer risk [30]. Conversely, certain metabolites, such as secondary bile acids and trimethylamine N-oxide (TMAO), have been implicated in processes associated with DNA damage, genomic instability, and angiogenesis, fostering an environment conducive to carcinogenesis [31,32]. Furthermore, the interplay between the gut microbiota, metabolites, and cancer therapies accentuates their potential role in modulating drug efficacy and patient outcomes [33,34].

The human gut microbiota is a highly diverse and interconnected ecosystem of trillions of microbial cells and thousands of microbial species. Its composition is influenced by various factors, including diet, age, genetics, and environmental exposures [35]. A balanced gut microbiota is associated with overall health, promoting digestion, immune system maturation, and protecting against pathogenic infections [36,37]. The gut microbiota also participates in the metabolism of dietary compounds and endogenous molecules, contributing to the production of bioactive metabolites that can affect host physiology [38]. Moreover, it has been implicated in modulating the efficacy and toxicity of numerous drugs, including chemotherapeutic agents, through its enzymatic activities and interactions with drug metabolites [39,40].

### 1.2. Cannabis Overview

Cannabis, belonging to the Cannabaceae family, has a rich history of diverse applications, encompassing medicinal, recreational, and industrial uses. Taxonomically, the genus Cannabis presents uncertainties, with three disputed species, *C. sativa*, *Cannabis indica*, and *Cannabis ruderalis,* leading to alternative classifications suggesting subspecies or a singular undivided species [41,42,43]. The plant is recognized as an annual, dioecious, flowering herb with palmately compound or digitated leaves, believed to be indigenous to Asia [44]. Industrially, the plant yields hemp fibre for diverse applications, including paper, clothing, biofuel, and food products [45,46]. The chemical composition of cannabis is complex, comprising over 400 molecules, with approximately 100 identified as cannabinoids, alongside terpenoids, flavonoids, and essential fatty acids [47,48]. Classifying the cannabis plant can be difficult because of the variations in its appearance, both in the wild and in cultivation. Thus, this can create uncertainties in the plant’s classification system [49]. Additionally, the two most researched compounds found in cannabis are THC and CBD [50]. THC is responsible for the psychotropic effects of cannabis, while CBD is non-intoxicating and possesses various therapeutic properties, including anti-inflammatory, analgesic, and anxiolytic effects [51,52]. The ECS is a key player in mediating the effects of cannabinoids in the human body. Cannabinoid receptors, CB1 and CB2, are a part of the ECS and widely distributed throughout the body, including the gastrointestinal tract, and regulate various physiological functions, such as pain perception, appetite, and immune responses [16,53]. Preclinical and clinical studies have demonstrated that cannabinoids exhibit anticancer properties by inducing apoptosis, inhibiting cell proliferation, and inhibiting angiogenesis in various cancer types [54,55]. Moreover, cannabis has shown promise in alleviating cancer and chemotherapy-related symptoms such as pain, nausea, and loss of appetite in people living with cancer [56].

## 2. Cannabis in Cancer Research

Cannabis has attracted significant attention in recent years for its potential anticancer properties. Preclinical studies using various cancer cell lines and animal models have demonstrated that cannabinoids, such as THC and CBD, possess antitumor effects through multiple mechanisms [54,55]. These mechanisms include induction of apoptosis, inhibition of cell proliferation, suppression of angiogenesis (blood vessel formation supporting tumour growth), and prevention of metastasis [55]. Furthermore, cannabinoids have clinically shown promise in alleviating cancer-related symptoms and improving the quality of life of people with cancer. For instance, cannabis-based products have been used to manage chemotherapy-induced nausea and vomiting, reduce cancer-related pain, and stimulate appetite in people living with cancer [56,57].

### 2.1. Anticancer Properties of Cannabinoids

#### 2.1.1. Induction of Apoptosis

Apoptosis is a crucial process for maintaining tissue homeostasis, and its dysregulation can contribute to cancer development and progression. Cannabinoids, particularly THC, have been shown to induce apoptosis in various cancer cell lines, including breast, prostate, lung, and colon cancer cells (Figure 1) [54,55]. The pro-apoptotic effects of cannabinoids are often mediated through the activation of cannabinoid receptors, especially CB1 and CB2, leading to the generation of reactive oxygen species and subsequent cell death signalling cascades [55,58].

Cannabinoids have been reported to suppress cancer cell proliferation by arresting the cell cycle and preventing cell division. In various cancer cell lines, such as MDA-MB-231 human breast cancer, prostate cancer (PC3, DU145, LNCaP), and prognostic colon cancer (Ki-67) cells, cannabinoids have demonstrated the ability to downregulate key cell cycle regulatory proteins and inhibit cell cycle progression, leading to reduced cell proliferation [55,59,60,61].

Alsherbiny et al. investigated the potential interactions between CBD and five commonly used chemotherapeutic drugs (docetaxel, doxorubicin, paclitaxel, vinorelbine, and 7-ethyl-10-hydroxycamptothecin) in MCF7 breast adenocarcinoma cells using multiple synergy quantitation models [62]. The authors observed a strong synergy between CBD and 7-ethyl-10-hydroxycamptothecin (SN-38) and vinorelbine (VIN) at various molar ratios [62]. Interestingly, synergy was observed between CBD and all the chemotherapeutic drugs when combined at a specific molar ratio of 636:1. However, some discordant synergy trends were noted, necessitating further validation [62]. The study found that combinations of CBD with chemotherapeutic drugs enhanced apoptosis in MCF7 cells compared to individual drug treatments or the negative control. A shotgun proteomics analysis revealed 121 dysregulated proteins in CBD-treated MCF7 cells compared to the negative control [62]. Notably, the authors identified previously unreported cytotoxic mechanisms of CBD, including the potential inhibition of topoisomerase II β and α, cullin 1, V-type proton ATPase, and CDK-6, along with disrupted energy production and reduced mitochondrial translation. When CBD was combined with SN-38 (CSN-38), 91 significantly dysregulated proteins were identified [62]. These proteins were associated with key molecular pathways such as telomerase regulation, cell cycle control, topoisomerase I function, EGFR1 signalling, protein metabolism, TP53-mediated DNA repair, death receptor signalling, and RHO GTPase signalling. These pathways were potentially attributed to the synergistic effects observed between CBD and SN-38 [62].

#### 2.1.2. Suppression of Angiogenesis

Angiogenesis, the formation of new blood vessels, is essential for tumour growth and metastasis. Cannabinoids, particularly CBD, have been shown to inhibit angiogenesis in both in vitro and in vivo studies. Notably, CBD can modulate various signalling pathways involved in angiogenesis, including the vascular endothelial growth factor (VEGF) pathway, thus reducing the formation of new blood vessels in tumours [63,64].

#### 2.1.3. Prevention of Metastasis

Metastasis is a hallmark of cancer and a major contributor to cancer-related mortality. Several studies have indicated that cannabinoids can inhibit metastatic processes, including tumour cell migration, invasion, and adhesion, such as in the highly invasive HeLa cervical cancer, human cervical (C33A), lung carcinoma (A549), and prostate cancer cell lines [65,66]. The downregulation of matrix metalloproteinases (MMPs) and the modulation of cell adhesion molecules were reported mechanisms by which cannabinoids exert their anti-metastatic effects [67].

### 2.2. Potential Role of Gut Microbiota in Modulating the Anticancer Effects of Cannabis

Emerging evidence suggests that the gut microbiota could play a critical role in modulating the anti-cancer effects of cannabis. As aforementioned, gut microbiota can metabolize cannabinoids, potentially influencing their pharmacological activity and bioavailability. Moreover, gut microorganisms produce various metabolites that can interact with the ECS and affect its regulation [68]. Consequently, the composition and function of gut microbiota may impact the local concentration and activity of cannabinoids in the gastrointestinal tract, where gut cancers often originate. Furthermore, the gut microbiota has been linked to systemic immune responses and inflammation, which can influence the tumour microenvironment and cancer progression [69,70]. It is conceivable that alterations in the gut microbiota composition could affect the immune response to cannabinoids and, consequently, their anticancer effects.

### 2.3. Implications for Cancer Treatment

Understanding the influence of gut microbiota on cannabis pharmacology has significant implications for cancer treatment. The ability of the gut microbiota to metabolize cannabinoids and influence their pharmacological activity could impact the anti-cancer properties of cannabis in the gastrointestinal tract and systemically. Additionally, given the influence of gut microbiota on cannabinoid receptor activation and the ECS, gut microbiota-targeted interventions may offer a strategy to enhance the therapeutic effects of cannabinoids in cancer treatment. By modulating the gut microbiota through dietary interventions, probiotics, or prebiotics, it might be possible to potentiate the antitumor effects of cannabinoids or mitigate potential side effects associated with chronic cannabis use. Furthermore, the gut microbiota has been implicated in modulating the response to cancer immunotherapies [69,70]. As cannabinoids also exhibited immune-modulatory properties, gut microbiota interactions with cannabinoids could potentially impact the response to cannabis-based immunotherapies in cancer.

The gut microbiota has recently emerged as a crucial player in modulating the pharmacological effects of various therapeutic agents, including chemotherapeutic drugs [71,72]. By interacting with drugs and their metabolites, the gut microbiota can influence drug bioavailability, metabolism, and efficacy [71,72,73]. Moreover, the gut microbial metabolites, such SCFAs, interact with the ECS and influence its regulation [68]. SCFAs have been shown to modulate the expression and activity of cannabinoid receptors, affecting various physiological processes in the gut, including inflammation and motility [74]. As inflammation and gut dysmotility are implicated in cancer development and progression, the ability of gut microbiota to produce SCFAs may play a role in modulating the anticancer effects of cannabinoids. In particular, the gut microbiota may impact the anticancer effects of cannabis by affecting the bioactivation, metabolism, and local concentrations of cannabinoids in the gastrointestinal tract.

## 3. Gut Microbiome and Cannabis Interactions

### 3.1. The ECS and Gut Microbiota

The ECS regulates various physiological functions in the gastrointestinal tract, such as gut motility, secretion, and visceral sensation. Activation of CB1 receptors in the enteric nervous system can inhibit neurotransmitter release, resulting in reduced gastrointestinal motility and intestinal secretion (Figure 2) [75,76]. On the other hand, CB2 receptor activation in immune cells within the gut mucosa can influence the inflammatory response and contribute to gut immune homeostasis [77].

Interestingly, the gut microbiota has been shown to interact with the ECS, further highlighting the relationship between the gut microbiota and cannabis. Interestingly, germ-free mice, which lack gut microbiota, displayed alterations in the expression of ECS components in the gastrointestinal tract compared to conventionally raised mice [77]. These findings suggested that gut microorganisms are essential for optimal ECS regulation in the gut. Furthermore, the interaction between the ECS and the gut microbiota is bidirectional, as the ECS can influence the gut microbiota composition and function. For example, activation of CB1 receptors has been shown to affect the gut microbiota by altering the gut transit time and promoting changes in gut motility [78]. Studies have also suggested that the ECS might play a role in modulating gut barrier integrity and immune responses, which can influence the gut microbiota [77,79]. Conversely, the gut microbiota can produce endocannabinoids and influence the endocannabinoid tone in the gut. For instance, commensal bacteria have been found to produce endocannabinoid-like compounds, such as 2-arachidonoyl glycerol (2-AG), which can activate cannabinoid receptors and modulate gut functions [80]. Moreover, the gut microbiota can guide the expression and activity of endocannabinoid-degrading enzymes, thereby regulating endocannabinoid levels locally [16]. The crosstalk between the ECS and gut microbiota has significant implications for gut health and overall physiology. Dysregulation of the ECS in the gut has been associated with various gastrointestinal disorders, including inflammatory bowel disease (IBD), irritable bowel syndrome (IBS), and colorectal cancer [81,82]. Additionally, alterations in the gut microbiota composition have been linked to ECS dysfunction and associated gut pathologies [68,75,83].

Understanding the interactions between the ECS and gut microbiota is of growing interest due to their potential implications for human health and disease. As the use of cannabis-based products for medical purposes becomes more prevalent, it is crucial to investigate how cannabis interacts with the gut microbiota and how these interactions influence various physiological processes, including those relevant to cancer development and treatment.

### 3.2. Impact of Cannabis Administration on Gut Microbiota Composition

Emerging evidence suggests that cannabis consumption can alter the gut microbiota composition. Animal studies have demonstrated changes in the abundance of specific bacterial taxa following cannabis exposure [13,84]. Moreover, clinical studies have shown associations between cannabis use and shifts in the gut microbial community, particularly in relation to the abundance of certain bacterial communities [13,85]. A recent study by Al-Ghezi et al. (2019) reported that chronic THC administration in mice led to changes in the gut microbiota, with an increase in the relative abundance of *Akkermansia muciniphila*, a bacterium associated with improved gut barrier function and metabolic health. Conversely, cannabis use has been linked to an increased abundance of *Bacteroides* species in the human gut microbiota, which might be associated with gut inflammation and metabolic disorders [85]. These findings suggest that cannabis consumption can influence gut microbiota composition, potentially affecting gut health and various physiological processes regulated by the gut microbiome.

### 3.3. Gut Microbiota-Mediated Metabolism of Cannabinoids

The gut microbiota can play a crucial role in the metabolism of cannabinoids, affecting their bioavailability and pharmacological activity. Several studies have identified microbial enzymes capable of metabolizing cannabinoids, leading to the formation of active or inactive metabolites [86]. For instance, the gut microbiota possesses the enzyme β-glucuronidase, which can deconjugate glucuronide metabolites of THC, thus releasing the active form of THC back into circulation [39]. The gut microbiota can also influence systemic exposure to cannabinoids by altering the enterohepatic circulation [87]. Bile acids, metabolites produced by the liver and modified by the gut microbiota, can form complexes with cannabinoids, leading to their reabsorption and prolonged retention within the body [11]. The capacity of gut microbiota to metabolize cannabinoids can significantly impact their bioavailability and therapeutic effects, further highlighting the crosstalk between the gut microbiota and cannabis in modulating physiological responses.

### 3.4. Influence of Gut Microbiota on Cannabis Pharmacology

The complex interactions between cannabinoids and gut microbes have the potential to impact the therapeutic efficacy and safety of cannabis-based treatments for various medical conditions, including cancer.

One of the key mechanisms by which the gut microbiota influences cannabis pharmacology is the metabolism of cannabinoids. Several studies have demonstrated that gut bacteria possess the enzymatic machinery necessary to metabolize cannabinoids into different compounds, which can significantly alter their pharmacological activity [11]. For example, ΔTHC, the primary psychoactive compound in cannabis, is metabolized by gut bacteria into 11-hydroxy-THC (11-OH-THC) and 11-nor-9-carboxy-THC (THC-COOH) [88]. These metabolites exhibit distinct pharmacological properties compared to parent THC, with 11-OH-THC reported to be more potent than THC, while THC-COOH is considered inactive but is often used as a plasma marker of cannabis consumption in drug tests [51,88]. Additionally, certain gut bacteria can metabolize CBD into 7-hydroxy-CBD, a compound with potential anti-inflammatory properties [89].

### 3.5. Modulation of Cannabinoid Receptor Activation

Alterations in the gut microbiota have been associated with changes in endocannabinoid levels and expression of cannabinoid receptors in the gut and the brain, potentially impacting various physiological processes [83,90]. Influencing cannabinoid receptor activation and, in turn, affecting the overall pharmacological response to cannabinoids is another key role of gut microbiota. Cannabinoids primarily exert their effects by binding to and activating cannabinoid receptors, particularly CB1 and CB2 (Figure 3) [55]. Preclinical studies have shown that certain gut bacteria-derived metabolites, such as secondary bile acids, can activate cannabinoid receptors (Figure 3) [91]. These secondary bile acids are produced through the gut microbiota-mediated deconjugation of primary bile acids, which are synthesized in the liver and secreted into the intestine [91].

### 3.6. Gut Microbiota and Cannabinoid-Induced Gut Motility

Cannabinoids have well-known effects on gut motility, influencing gastrointestinal transit and smooth muscle contractions [76]. The gut microbiota can modulate these effects by interacting with the ECS and affecting the production of certain neurotransmitters and neuromodulators involved in gut motility regulation. For instance, gut bacteria can produce neurotransmitters like serotonin, dopamine, and gamma-aminobutyric acid (GABA) [92]. These neurotransmitters can interact with cannabinoid receptors and modulate the effects of cannabinoids on gut motility and gastrointestinal functions [92].

Recent investigations have revealed a growing connection between gut microbiota and conditions related to metabolism and the nervous system. Observational studies have indicated that certain dietary interventions involving specific fatty acids may lead to increased levels of endocannabinoids (eCBs). These changes in eCBs have been linked to variations in microbial families such as *Peptostreptococcaceae*, *Veillonellaceae*, and *Akkermansiaceae* [14]. Additionally, the use of cannabis has been associated with alterations in eCB levels, promoting mucosal healing in ulcerative colitis patients and enhancing their overall quality of life [14]. The manipulation of the ECS through cannabinoids has also shown promise in immune suppression. For instance, recent studies have demonstrated that endocannabinoid anandamide (AEA) can counteract harmful microbiota disruptions caused by severe acute respiratory distress syndrome (ARDS) in mice [14]. AEA treatment was associated with increased levels of beneficial bacteria producing SCFAs, such as butyrate, which play a crucial role in gut health [14]. This treatment also reduced inflammation of lung and gut-associated lymph nodes, bolstering epithelial barrier integrity. Moreover, with cannabinoid treatment, pathogenic bacteria were mitigated, and the presence of beneficial bacteria was enhanced [14].

Similar investigations have highlighted the impact of cannabinoids on diseases like colitis. In a murine colitis model, activating cannabinoid receptors by THC improved the integrity of the colonic barrier and enhanced the production of protective molecules. While alterations in the gut microbiota were observed, they were not directly linked to the positive effects of THC. Another study demonstrated the synergistic effects of fish oil and CBD in treating colitis in mice. This combination therapy reduced inflammation markers and improved intestinal permeability, suggesting a potential avenue for managing gastrointestinal diseases.

Synthetic cannabinoids have been extensively studied for their anti-inflammatory properties and potential to influence the gut microbiota. For instance, the CB2R agonist JWH133 demonstrated benefits in cirrhotic rats by reducing bacterial overgrowth and promoting intestinal integrity [14]. Conversely, blocking the CB1 receptor with the anti-obesity drug Rimonabant led to reductions in obesity, inflammatory cytokines, and certain gut bacterial strains [14]. Clinical investigations have also shown positive outcomes with synthetic cannabinoids, such as nabilone, in alleviating symptoms like post-traumatic stress disorder and diarrhoea [93].

In summary, ongoing research emphasizes the interactions between cannabinoids, gut microbiota, and various health conditions. Findings suggest that cannabinoids, whether natural or synthetic, hold promise in modulating gut microbiota, influencing inflammation, and improving conditions related to metabolism and the nervous system. These cannabis-induced changes in the gut microbiota may, in turn, influence the effects of cannabinoids on gut motility and other gut-related functions.

## 4. Gut Microbiome, Cannabis, and Cancer: Potential Interplay

### 4.1. Gut Microbiome-Mediated Modulation of Immune Responses

The gut microbiome profoundly impacts the development and function of the immune system, including immune cell development, shaping the composition of immune cell populations, and modulating the production of inflammatory mediators [37]. Consequently, alterations in the gut microbiome composition have been linked to various immune-related diseases, including cancer [69]. Furthermore, the gut microbiota can shape systemic immune responses and influence immune cell trafficking to distant sites, including tumours [70]. Alterations in the gut microbiota have been associated with changes in the tumour microenvironment and the response to cancer immunotherapy [69,70]. Hence, the gut microbiota may modulate the response to cannabinoids in cancer therapy by influencing the immune microenvironment of tumours.

### 4.2. The Bidirectional Relationship: Cannabis, Gut Microbiota, and Cancer

The interaction between cannabis, the gut microbiome, and cancer is likely bidirectional, with each component influencing the others in a complex and dynamic manner (Figure 4). Cannabis consumption may impact the gut microbiota composition and function by directly affecting gut motility, secretions, and immune responses. In turn, the gut microbiota can metabolize cannabinoids, affect their pharmacological activity, and modulate the immune response to cannabinoids in the gastrointestinal tract and systemically. Dysregulation of the gut microbiota has been associated with gastrointestinal cancers and other malignancies, highlighting the importance of understanding how cannabis interactions with the gut microbiota might impact cancer development and progression. However, much remains to be elucidated regarding the specific mechanisms and clinical implications of the interactions between cannabis, the gut microbiota, and cancer. Further research is needed to unravel the complexity of these interactions and identify potential therapeutic strategies to harness the gut microbiota–cannabis axis for cancer management.

### 4.3. Modulation of the Gut Microbiota to Potentially Enhance the Effects of Cannabis in Cancer Therapy

#### Cancer Pain and Chemotherapy Side Effects Management

Johnson et al. (2010) examined the use of THC: CBD extract in people with cancer who experienced intractable pain. The most common side effects reported were dizziness and dry mouth, and the authors noted that these side effects were generally mild to moderate in severity [56]. Portenoy et al. (2012) also investigated a placebo-controlled trial that assessed the use of a novel cannabinoid formulation called nabiximols (Sativex) in people with cancer experiencing poorly controlled chronic pain. The side effects included dizziness, fatigue, and dry mouth, with no significant differences in side effects between the treatment and placebo groups [94]. Another observational study by Bar-Sela et al. (2013) examined the use of medicinal cannabis in people with cancer receiving supportive or palliative care, and the reported side effects included dizziness, dry mouth, and psychoactive effects with most being mild to moderate and did not lead to treatment discontinuation [95]. Another report by the National Academies of Sciences, Engineering, and Medicine acknowledged potential side effects such as dizziness, dry mouth, and cognitive impairment, based on the current evidence of cannabis and cannabinoids use for various medical conditions, including cancer-related pain, emphasizing the importance of further research to understand these side effects [96]. These studies collectively suggested that while cannabis and its derivatives may benefit from managing cancer-related pain, they can be associated with side effects such as dizziness, dry mouth, and fatigue.

Strategies to modulate the gut microbiota, such as prebiotics, probiotics, and dietary interventions, could potentially reduce the side effects and enhance the therapeutic effects of cannabis in cancer treatment. Preclinical studies have shown that certain probiotics and dietary components can influence cannabinoid receptor expression, endocannabinoid levels, and gut barrier integrity [83,97]. Therefore, combining cannabis-based treatments with gut microbiome-targeted interventions may provide synergistic effects, improving the therapeutic properties of cannabinoids. Moreover, gut microbiota modulation could be explored to reduce potential adverse effects of cannabis use, such as gastrointestinal discomfort and altered gut motility. By promoting gut health and intestinal homeostasis, interventions targeting the gut microbiota could potentially improve the tolerability and overall safety of cannabis-based therapies.

## 5. Safety Considerations and Clinical Trials

Cannabis-based therapies have emerged as a promising avenue for cancer treatment, primarily due to their potential to alleviate cancer symptoms and inhibit tumour growth [98,99]. However, alongside these promises, it is imperative to rigorously assess the safety considerations associated with these therapies. The safety considerations of cannabis-based therapies are multifaceted. Upholding ethical principles, such as patient autonomy, informed consent, and privacy, is paramount in pursuing cannabis-based therapies for cancer treatment.

One crucial aspect that warrants in-depth investigation is the role of the gut microbiota, as it can metabolize cannabinoids into active and potentially toxic compounds [100], which could influence the safety and efficacy of cannabis-based therapies. As aforementioned, the gut microbiota can convert THC into its more psychoactive form, 11-hydroxy-THC, which may lead to unwanted psychotropic effects and safety concerns [9]. In clinical trials exploring cannabis-based therapies for cancer treatment, understanding how gut microbiota composition influences cannabinoid metabolism and toxicity is paramount [101].

### Personalized Medicine and Gut Microbiota Profiling

The gut microbiota exhibits considerable interindividual variation, influenced by genetics, diet, lifestyle, and exposure to various environmental factors [102]. Individual gut microbiome variations could result in disparate treatment responses and safety profiles among patients. This variability in the gut microbiota may result in diverse responses to cannabis-based treatments in people with cancer. Therefore, it is crucial to consider the gut microbiota as a potential modifier of treatment outcomes [9]. Future clinical trials in this field could benefit from incorporating gut microbiota profiling as part of their study design. By characterizing the gut microbiome of participants before and during treatment, researchers can gain insights into how specific microbiota profiles may correlate with treatment response and safety. This personalized approach may facilitate the identification of patient subgroups more likely to respond positively to cannabis-based therapies and help assess how gut microbiota alterations might influence treatment outcomes.

The relationship between the gut microbiota and the effects of cannabis has been a subject of growing research interest, with potential implications for individual variations in cannabinoid metabolism, immune response, and gut barrier integrity. Here are some instances where this relationship has been explored. A study found that THC administration enriched short-chain fatty acids (SCFAs), specifically propionic acid, which attenuated the inflammatory response and protected mice from fatality [14]. Also, there is a hypothesis that the crosstalk between the microbiota and the intestinal endocannabinoid system plays a prominent role in stress-induced changes in metabolism [103]. A systematic review and meta-analysis revealed associations between cannabis usage and alterations in the human microbiome, which must be considered in future research on the therapeutic effects of cannabis on patients [104]. Cannabis extracts increased microbial diversity and richness in a mouse model of metabolic disease while promoting the enrichment of microbial taxa associated with health [105].

Moreover, research has shown that cannabinoids modulate the microbiota-gut-brain axis in HIV/SIV infection by reducing neuroinflammation and dysbiosis while concurrently elevating endocannabinoid levels [106]. Researchers are exploring how cannabis smoking alters the bacterial communities in the mouth and its potential impacts on the brain [107]. These examples demonstrate the potential impact of the gut microbiota on the effects of cannabis and the need for further research to understand how this relationship could guide the selection of appropriate cannabis-based formulations, dosages, and treatment regimens tailored to individual gut microbiota composition and characteristics.

By profiling the gut microbiota, healthcare professionals may also gain insights into individual variations in cannabinoid metabolism, immune response, and gut barrier integrity. This information could guide the selection of appropriate cannabis-based formulations, dosages, and treatment regimens tailored to everyone’s unique gut microbiota composition and characteristics.

## 6. Conclusions and Future Directions

In conclusion, the emerging field of research on the gut microbiome–cannabis axis presents a promising avenue for advancing cancer treatment strategies. To fully unlock the therapeutic potential of this axis, future investigations should prioritise several key research directions. This includes understanding how specific microbial species metabolise cannabinoids, influencing their bioavailability and therapeutic actions. Additionally, exploring the interplay between the gut microbiota, the immune system, and cannabis can unravel how these interactions impact the immune response against cancer. Investigating the role of the gut microbiota in maintaining gut barrier integrity and its influence on the effects of cannabis is also crucial for optimising treatment efficacy.

Despite significant advancement in the field, limitations and unanswered questions persist. The complexity and variability of the gut microbiota make it challenging to establish universal guidelines for cannabis-based cancer treatments. Large-scale studies are needed to identify specific gut microbiome signatures associated with optimal treatment responses, enabling the selection of patients more likely to benefit from cannabis-based therapies and optimising treatment outcomes. Large-scale longitudinal studies and clinical trials incorporating gut microbiome profiling are essential to elucidate the complex mechanisms and clinical implications of the gut microbiome-cannabis axis in cancer therapy. Gut microbiome profiling offers the potential to identify patient-specific microbiome signatures that influence treatment responses to cannabis-based therapies, optimising outcomes and minimising adverse effects. Specific aspects, such as the precise taxonomic composition of bacterial species involved, the molecular mechanisms at play, and the clinical implications for cancer prevention and treatment, also warrant further empirical investigation. Additionally, the long-term safety and efficacy of gut microbiome-targeted interventions in combination with cannabis-based therapies require thorough investigation. These studies are necessary for developing personalised therapeutic approaches while upholding ethical considerations, especially related to the safety and side effects of cannabis in people with cancer.

Advancing our understanding of the gut microbiome–cannabis axis for cancer treatment necessitates multidisciplinary collaborations among researchers and healthcare professionals, including microbiologists, oncologists, pharmacologists, and immunologists. This collaborative approach will ensure a comprehensive understanding of the underlying mechanisms and expedite the translation of research findings into clinical applications.

In summary, the crosstalk between the gut microbiota and cannabis in relation to cancer represents a promising and emerging area of cancer therapy. While this field is still in its infancy, evidence from preclinical studies suggests that the gut microbiota plays a crucial role in modulating the pharmacological effects of cannabinoids, influencing their metabolism, immune-modulatory properties, and interaction with the gut barrier. Moreover, the gut microbiota and its metabolites have been implicated in cancer development and progression, indicating its potential as a target for therapeutic interventions in combination with cannabis-based therapies. Combining cannabis-based therapies with strategies to modulate the gut microbiome, such as probiotics, prebiotics, or dietary interventions, holds promise for enhancing the anticancer properties of cannabinoids. While challenges and limitations exist, addressing these research directions will contribute to realising the full potential of the gut microbiota–cannabis–cancer axis.

## Figures and Tables

**Figure 1 ijms-25-00872-f001:**
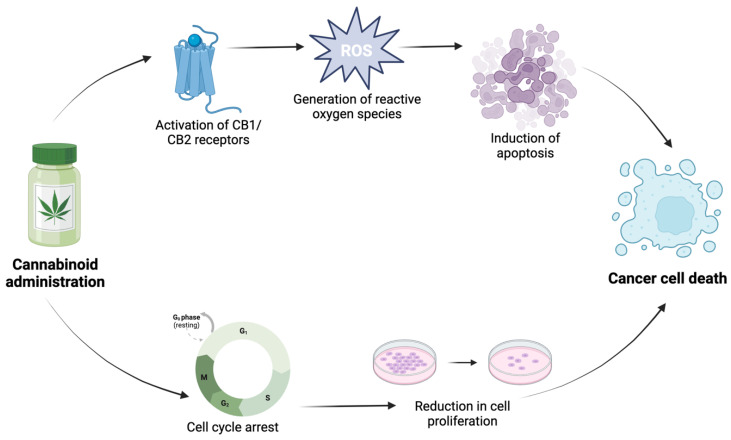
Effect of cannabinoid administration on cancer cell death, including the induction of apoptotic cell death and a reduction in cell proliferation.

**Figure 2 ijms-25-00872-f002:**
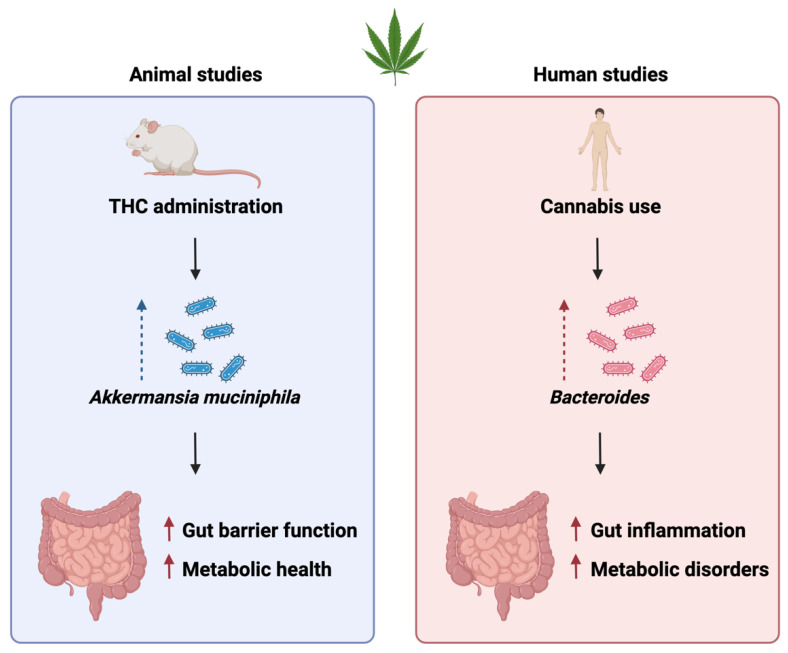
A diagrammatic breakdown of animal and human studies examining cannabis-related use and the impact on gut physiological function.

**Figure 3 ijms-25-00872-f003:**
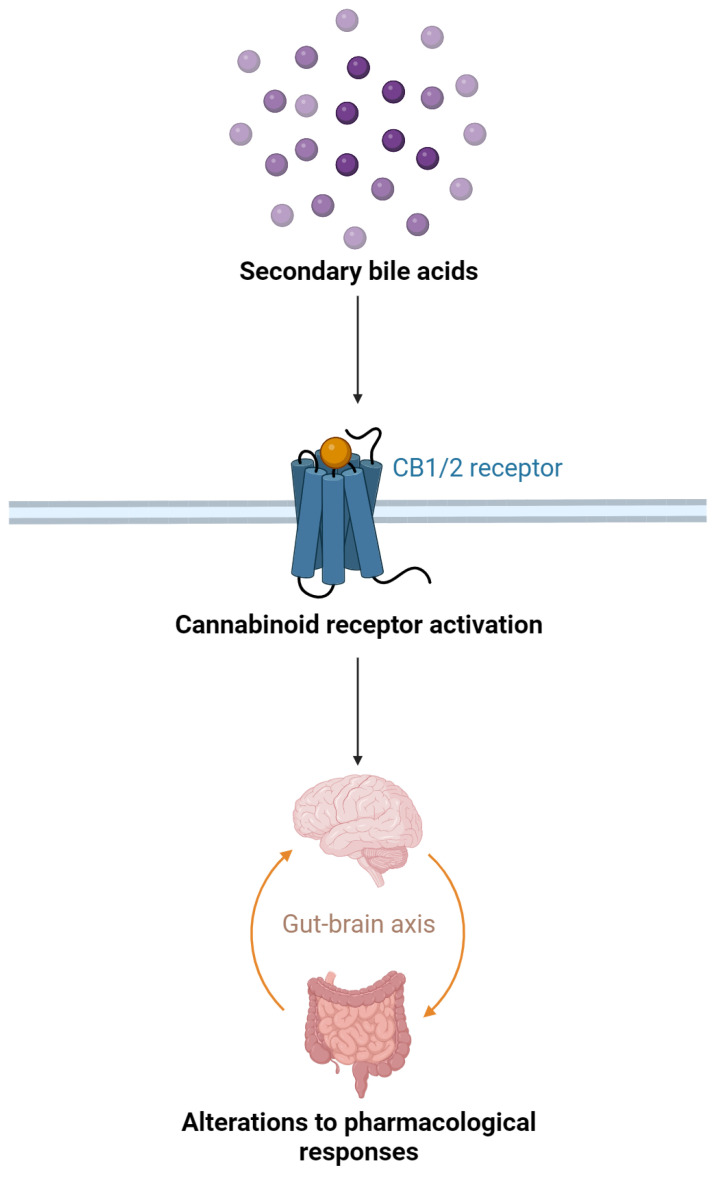
The mechanisms of cannabinoid receptor activation by secondary bile acids.

**Figure 4 ijms-25-00872-f004:**
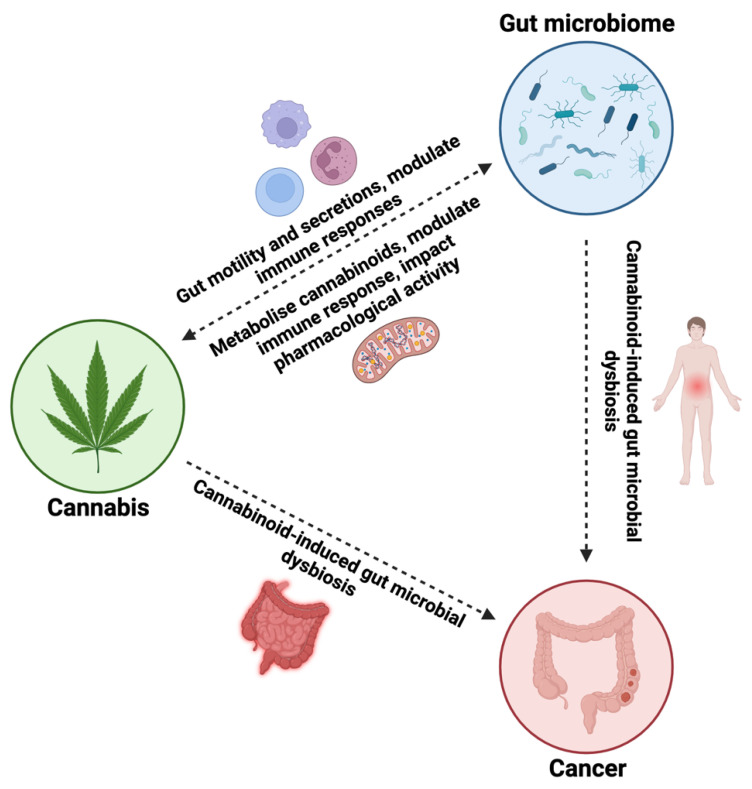
Bidirectional association between cannabis use, gut microbial composition, and cancer onset, including the potential metabolic pathways involved and physiological effects.

## Data Availability

No new data were created or analyzed in this study. Data sharing is not applicable to this article.

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
