# Peer review of "Buds and Bugs: A Fascinating Tale of Gut Microbiota and Cannabis in the Fight against Cancer"

_ijms, 2024, doi:10.3390/ijms25020872_

Round 1

Reviewer 1 Report

Comments and Suggestions for Authors

Abstract section is written clearly and focused. The strength of this article is that it presents a clear summation of a particular issue. The Authors describe an interesting review that focuses on the correlation between the gut microbiome and cannabis, as well as on the novel perspectives in pharmacotherapy of oncology diseases. Additionally, the Authors cited the newest available research (e.g. from 2022). However, in some paragraphs of the manuscript, the text is too a detailed what causes that it is incomprehensible. I suggest that the Authors should avoid unnecessary information.

Besides, there are a few minor issues that need to be addressed before it is publishable. I have few comments that may be helpful to improve the manuscript. My concerns are as follows:

·         In the manuscript, there is a lot of abbreviations. Thus, it seems to me that a list of abbreviations should be added at the beginning of the manuscript.

·         The titles of the Figures should be more clearly described. The title of Figure 4. : “the mechanisms of cannabinoid receptor activation” should be changed to:  “The mechanisms of cannabinoid receptor activation”. 

·         Next: Figure 4. Bidirectional association between cannabis use, gut microbial composition, and cancer onset, including the potential metabolic pathways involved and physiological effects” should be changed to Figure 5 (and the same in text of manuscript – page 12, line 423).

In my opinion the current version of the manuscript entitled “Buds and Bugs: A Fascinating Tale of Gut Microbiota and Cannabis in the Fight Against Cancer” written by Ahmad K. Al-Khazaleh, Kayla Jaye, Dennis Chang, Gerald W. Münch and Deep Jyoti Bhuyan is good and could be publish in the Journal: IJMS after minor revision.

Comments on the Quality of English Language

Abstract section is written clearly and focused. The strength of this article is that it presents a clear summation of a particular issue. The Authors describe an interesting review that focuses on the correlation between the gut microbiome and cannabis, as well as on the novel perspectives in pharmacotherapy of oncology diseases. Additionally, the Authors cited the newest available research (e.g. from 2022). However, in some paragraphs of the manuscript, the text is too a detailed what causes that it is incomprehensible. I suggest that the Authors should avoid unnecessary information.

Besides, there are a few minor issues that need to be addressed before it is publishable. I have few comments that may be helpful to improve the manuscript. My concerns are as follows:

·         In the manuscript, there is a lot of abbreviations. Thus, it seems to me that a list of abbreviations should be added at the beginning of the manuscript.

·         The titles of the Figures should be more clearly described. The title of Figure 4. : “the mechanisms of cannabinoid receptor activation” should be changed to:  “The mechanisms of cannabinoid receptor activation”. 

·         Next: Figure 4. Bidirectional association between cannabis use, gut microbial composition, and cancer onset, including the potential metabolic pathways involved and physiological effects” should be changed to Figure 5 (and the same in text of manuscript – page 12, line 423).

In my opinion the current version of the manuscript entitled “Buds and Bugs: A Fascinating Tale of Gut Microbiota and Cannabis in the Fight Against Cancer” written by Ahmad K. Al-Khazaleh, Kayla Jaye, Dennis Chang, Gerald W. Münch and Deep Jyoti Bhuyan is good and could be publish in the Journal: IJMS after minor revision.

Author Response

General comments

Abstract section is written clearly and focused. The strength of this article is that it presents a clear summation of a particular issue. The Authors describe an interesting review that focuses on the correlation between the gut microbiome and cannabis, as well as on the novel perspectives in pharmacotherapy of oncology diseases. Additionally, the Authors cited the newest available research (e.g. from 2022). However, in some paragraphs of the manuscript, the text is too a detailed what causes that it is incomprehensible. I suggest that the Authors should avoid unnecessary information.

We thank the reviewer for their positive feedback. However, we think that sometimes detailed information is necessary, especially for non-specialised readers who are new to the fields of gut microbiome, oncology and natural product chemistry.

Specific comments

Besides, there are a few minor issues that need to be addressed before it is publishable. I have few comments that may be helpful to improve the manuscript. My concerns are as follows:

  • In the manuscript, there is a lot of abbreviations. Thus, it seems to me that a list of abbreviations should be added at the beginning of the manuscript.

We appreciate the valuable feedback from the Reviewer. We have now added a list of abbreviations to the text highlighted in red, line (28-47).

  • The titles of the Figures should be more clearly described. The title of Figure 4. : “the mechanisms of cannabinoid receptor activation” should be changed to: “The mechanisms of cannabinoid receptor activation”.

Please note that Figure 4 is now Figure 3. We have now updated the title of Figure 3 to “The mechanisms of cannabinoid receptor activation by secondary bile acids” highlighted in red line (356).

  • Next: Figure 4. Bidirectional association between cannabis use, gut microbial composition, and cancer onset, including the potential metabolic pathways involved and physiological effects” should be changed to Figure 5 (and the same in text of manuscript – page 12, line 423).

Please note that the figure numbers are now updated. We thank the Reviewer for the valuable feedback.

In my opinion the current version of the manuscript entitled “Buds and Bugs: A Fascinating Tale of Gut Microbiota and Cannabis in the Fight Against Cancer” written by Ahmad K. Al-Khazaleh, Kayla Jaye, Dennis Chang, Gerald W. Münch and Deep Jyoti Bhuyan is good and could be publish in the Journal: IJMS after minor revision.

Reviewer 2 Report

Comments and Suggestions for Authors

Page 2, line 52: The term "symbiosis" does not seem correct. It may be changed to "interplay" or "interacción".

 Page 4, line 136: In the sentence "Moreover, THC and CBD are the most extensively studied," you refer to cannabinoids, but the previous sentence refers to the taxonomy of Cannabis. Please rephrase the paragraph.

 Figure 1: It is not clear why the gut microbiome-cannabinoid axis is presented as a cycle. I cannot find the link or the effect of the activation of cannabinoid receptors on cannabinoids. I consider that it can be illustrated in another way, as I do not think it is a cycle. Also, Figure 1 is not referred to in the text.

Author Response

Page 2, line 52: The term "symbiosis" does not seem correct. It may be changed to "interplay" or "interacción".

We appreciate this feedback from the Reviewer, and we have now updated that to “interaction” highlighted in red, line (73).

 Page 3, line 124: In the sentence "Moreover, THC and CBD are the most extensively studied," you refer to cannabinoids, but the previous sentence refers to the taxonomy of Cannabis. Please rephrase the paragraph.

We appreciate the valuable feedback from the Reviewer. We have now rephrased the text highlighted in red, line (147-150).

 Figure 1: It is not clear why the gut microbiome-cannabinoid axis is presented as a cycle. I cannot find the link or the effect of the activation of cannabinoid receptors on cannabinoids. I consider that it can be illustrated in another way, as I do not think it is a cycle. Also, Figure 1 is not referred to in the text.

Figure 1 was an error and was supposed to be a graphical abstract. We have now renamed it as a graphical abstract and updated the figure as suggested by the reviewer. Also, we have updated the figure numbers in the manuscript accordingly.